# Boosting Targeted Black-Box Attacks via Ensemble Substitute Training and Linear Augmentation

**Xianfeng Gao [1], Yu-an Tan [1] , Hongwei Jiang [1], Quanxin Zhang [1] and Xiaohui Kuang [2,*]**

[1] School of Computer Science and Technology, Beijing Institute of Technology, Beijing 100081, China; ggxxff@bit.edu.cn (X.G.); tan2008@bit.edu.cn (Y.-a.T.); 2120171020@bit.edu.cn (H.J.); zhangqx@bit.edu.cn (Q.Z.)

[2] National Key Laboratory of Science and Technology on Information System Security, Beijing 100192, China

[*] Correspondence: 2120171093@bit.edu.cn

**Abstract:** These years, Deep Neural Networks (DNNs) have shown unprecedented performance in many areas. However, some recent studies revealed their vulnerability to small perturbations added on source inputs. Furthermore, we call the ways to generate these perturbations' adversarial attacks, which contain two types, black-box and white-box attacks, according to the adversaries' access to target models. In order to overcome the problem of black-box attackers' unreachabilities to the internals of target DNN, many researchers put forward a series of strategies. Previous works include a method of training a local substitute model for the target black-box model via Jacobian-based augmentation and then use the substitute model to craft adversarial examples using white-box methods. In this work, we improve the dataset augmentation to make the substitute models better fit the decision boundary of the target model. Unlike the previous work that just performed the non-targeted attack, we make it first to generate targeted adversarial examples via training substitute models. Moreover, to boost the targeted attacks, we apply the idea of ensemble attacks to the substitute training. Experiments on MNIST and GTSRB, two common datasets for image classification, demonstrate our effectiveness and efficiency of boosting a targeted black-box attack, and we finally attack the MNIST and GTSRB classifiers with the success rates of 97.7% and 92.8%.

**Keywords:** deep learning; adversarial attack; black-box attack; dataset augmentation; substitute training

## 1. Introduction

Deep Neural Networks (DNNs) have been widely used in many areas today, such as self-driving [1], speech recognition [2], image recognition and so on. In addition, in the image recognition field, DNN performs much more efficiently than other machine learning algorithms like Support Vector Machine (SVM) [3] or logistic regression [4], and has achieved great success in the ImageNet Large Scale Visual Recognition Challenge (ILSVRC) [5]. However, some people [6] found that, for an image recognition system, when the adversary adds tiny perturbations purposefully on a clean image, the classifier may misclassify the synthetic image, which looks the same as the source image, into any other desired class [7]. In addition, they call these purposefully synthetic images *Adversarial Examples* [8].

According to the knowledge of the target classifier, we can divide the approaches of generating adversarial examples into two types: white-box attack and black-box attack. The white-box attackers craft the adversarial examples based on the internal information of DNN, which might include the training dataset, outputs, hyper-parameters, gradients and feature maps. This makes this kind of attack easier than black-box attacks. The widely accepted white-box attack methods are Deep-Fool [9],

Jacobian-based Saliency Map Attack (JSMA) [10], Carlini and Wagner Attack (C&W) [11], Fast Gradient Sign Method (FGSM) [6], Iterative FGSM (I-FGSM) [12] and so on. For black-box attacks, we can only acquire the outputs responding to the paired inputs. This is a much stronger constraint for attackers. In addition, now there are One-Pixel Attacks [13], Boundary Attacks [14], and Jacobian-based Augmentation Attacks [15] well-known to adversarial attackers.

In order to overcome the problem that black-box adversaries have no access to the target classifiers' internal information, some researchers came up with the idea of training a local substitute model of the targeted model via the dataset augmentation method. Then, the attackers can craft adversarial examples via these substitute models using a white-box method but in a black-box assumption. Famously, Papernot et al. generated the substitute training and attacks by augmenting a few initial images using the Jacobian matrix of the substitute model combined with the pairs of the target classifier's inputs and outputs [15], which we called Jacobian-based Augmentation. They crafted adversarial examples using FGSM [6] and JSMA [10], and successfully misled the deep learning systems on Amazon and Google with high confidence. Moreover, Shi and Han [16] proposed a new dataset augmentation method named Schmidt Augmentation, and achieved a bit higher misclassification rate than the Jacobian-based Augmentation method. In addition, Xiao et al. [17] trained a substitute model and use its inside information for generating a black-box attack via GAN (Generative Adversarial Nets) [18], which we call advGAN.

Despite the high misclassification rate, there is no sign that the substitute models Papernot [15] or Yucheng [16] trained are able to generate targeted attacks. In order to perform high efficient targeted attacks, we improved the Jacobian-based augmentation by changing the augmenting dataset and reducing the information required from the Jacobian matrix to the single gradient value for every epoch. In this way which we named Gradient-Based Linear Dataset Augmentation, we successfully performed targeted attacks and largely decreased the times of querying black-box models. Moreover, to promote our success rate of black-box attacks, we think up the idea of ensemble attacks [19–21], which means that we train several substitute models respectively and generate adversarial examples by combining internal information of them. There is obvious elevation in success rates in this way. In addition, for solving the problem that the query times increase linearly with the number of substitute models, we proposed an Ensemble Training Method, which combines several models' gradient information and augments the dataset to train them in an ensemble. Finally, we achieved a very high success rate of targeted attacks with the same query number as one sub-model attack.

Therefore, our work makes great progress in the black-box attack of training substitute models and our contributions are the following:

- We propose a new dataset augmentation method that highly improves the accuracy of substitute models.
- We firstly make it generate targeted black-box attacks with high success rates via training substitute models.
- We propose an ensemble training method that trains multiple substitute models at a time, which boosts the targeted black-box attack without any extra query times.

The rest of our paper is organized as follows. We describe the threat model and introduce some notations in Section 2. Next, we will present some related works for adversarial attacks via training substitute models, especially the Jacobian-based augmentation. We also introduce the Carlini and Wagner Attack in Section 3 because of its relevance and importance to our works. In the next section, we propose our approaches of Gradient-Based Linear Dataset Augmentation and the Ensemble Training Method that mainly improve the effect of targeted attacks. After that, the experimental results with some comparisons will be presented in Section 5. Finally, we present the conclusions of our entire work.

## 2. Threat Model

With the popularity of DNNs on image classification, some researchers found that DNNs are vulnerable. A taxonomy of adversaries against DNN image classification is found in [10]. In our work, the adversaries attempt to find a small perturbation added on the input image that is not visible to human eyes but can force the DNN classifier to classify the input into any specific class. To achieve this, we consider a weak adversary with access to the DNN output only, which is called black-box assumption. In addition, in the context below, we will describe our threat model detailed and introduce some notations that may be frequently used in the following sections.

For an image classification system $O : x \rightarrow y$, it outputs a label $y$ or the predicting vector $\hat{y}$ corresponding to the input $x$. In adversarial attacks, our aim is to craft adversarial examples $x^* = x + \delta_x$, where $\delta_x$ is the perturbation. Corresponding to the adversarial goals, there are two kinds of adversarial examples: *non-targeted adversarial examples* and *targeted adversarial examples* [21]. As the names suggest, the difference between these two kinds of examples is whether the adversarial examples is just misclassified or classified to any other desired class $t$ by the target classifier.

Aiming at black-box attacks, we cannot acquire the inside parameters of target model $O$. In addition, here we consider targeted model outputs a probability vector $O(x)$ when given an input image $x$. In addition, $O(x)_i$ indicates the confidence of $O$ classifying $x$ into the component index $i$, which represents a specific class. Thus, we can easily know the predicted label, denoted as

$$O^*(x) = \arg\max(O(x)). \tag{1}$$

In addition, we consider that the adversary can only **query** $O$ to acquire the pair $(x, O(x))$. They do not have access to the internal information of the classifier $O$ or the training dataset. In addition, to make our attack closer to the real case, we assume that the adversary can only query the target model no more than 10% of its training dataset, which is are much stricter rules that many other works did not keep.

## 3. Related Work

Adversarial attacks have become a popular topic today due to the wide use of DNNs today. In addition, compared with white-box attack, black-box attacks attract more attention because they are much closer to realistic situations, which also means that they are more difficult.

### 3.1. Jacobian-Based Augmentation

Previously, Papernot et al. proposed the idea of Substitute Models Attack [15], which mainly includes training a local substitute model for the target black-box model via Jacobian-based augmentation and then using the substitute model to craft adversarial examples via white-box attack methods. This is the original and the most popular flowchart of generating substitute models attacks. In addition, in Figure 1, we describe the flowchart of this method in detail.

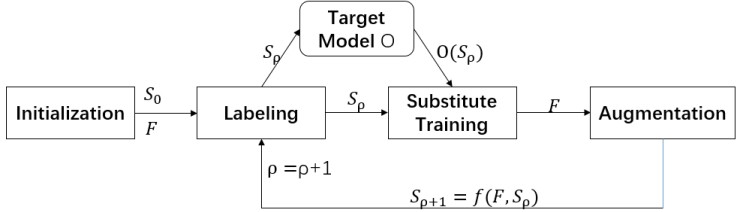

**Figure 1.** Flowchart of Substitute Training [15]. This pattern mainly contains four steps: Initialization, Labeling, Substitute Training and Augmentation. The adversaries label and augment the dataset repeatedly for training substitute models that fit the target model's decision boundary better and better. The crucial operation in this flowchart is Augmentation, which in other words is the definition of the augmenting function $f$.

For a target model $O$, we first choose the initial dataset $S_0$ and the architecture of substitute model $F$. In addition, then we continue to label the initial dataset $S_\rho$ so that we can train the substitute model $F$ with the training dataset $\{(x, O(x)) : x \in S_\rho\}$. After substitute training, we generate the augmentation based on dataset $S_\rho$ and the substitute model $F$ to get $S_{\rho+1}$, which we describe as

$$S_{\rho+1} = f(S_\rho, F), \tag{2}$$

where $f$ is the crucial element of substitute models' attack because it determines how we augment the dataset. Here in Jacobian-based augmentation, the augmenting function $f$ is constructed as

$$S_{\rho+1} = \{x + \lambda \cdot \mathrm{sgn}(J_F[O^*(x)]) : x \in S_\rho\} \cup S_\rho, \tag{3}$$

where $\lambda$ is a parameter of the augmentation: it defines the size of the step taken in the direction identified by the Jacobian matrix, and *sgn* is the sign function.

After the augmentation, we continue to label the new dataset $S_{\rho+1}$ like we proceeded with $S_\rho$ before, and train the substitute model $F$ with $\{(x, O(x)) : x \in S_{\rho+1}\}$. Here, we called $\rho$ *substitute training epoch*, denoted as *epoch $\rho$*. In addition, the method we present above is just a complete epoch, which we may repeat several times to finally get our substitute models. By the way, the number of substitute training epochs is mainly determined by the constraint of query times.

This is the main flowchart of Substitute Models Attack, and, in this paper, we follow this flowchart to train our substitute models. As the paper describes, the Jacobian-based Augmentation method acquired a high misclassification rate for both the local and the remote image classification system.

### 3.2. Carlini and Wagner Attack

Once the substitute models have been trained, the next step we take is to craft the adversarial examples. Thus, the methods for generating adversarial examples are very related to our works. Before introducing the method for crafting samples, we want to emphasize our adversarial goal first, which is different from the non-targeted attacks:

$$minimize\ \delta_x \qquad s.t.\ O(x + \delta_x) = t, \tag{4}$$

where $t$ is the target class to which we want this adversarial example to be classified.

In this paper, we mainly use the famous C&W [11] $L_2$ attack, which was proposed by Carlini, Nicholas and Wagner, and David in 2017. In addition, we will simply introduce this method in this following.

Given a source image $x$ and the target class $t$ (ensure that $t \neq O^*(x)$), we search for $z$ that solves

$$\underset{z}{minimize} \left\| \frac{1}{2}(\tanh(z) + 1) - x \right\|_2^2 + c \cdot f(\frac{1}{2}(\tanh(z) + 1)), \tag{5}$$

with $f$ defined as

$$f(x) = \max(\max\{O(x)_i : i \neq t\} - O(x)_t, -\mathcal{K}). \tag{6}$$

This $f$ is based on the best objective function found earlier, and by adjusting $\mathcal{K}$ we can adjust the confidence that target model $O$ has to classify our adversarial examples to target class $t$. The bigger $\mathcal{K}$ is, the higher confidence $O$ has to classify the adversarial example into a target class or misclassify the adversarial examples, but, on the other hand, the perturbation and the speed may be worse.

Notice that here $z$ does not mean the real perturbation while it has the same changing trend with the perturbation. Authors propose $z$ and *tanh* function here to guarantee the box constraint of the each pixel of the real adversarial samples $x^*$, which, in Equation (5), is denoted as $\frac{1}{2}(\tanh(z) + 1)$. In addition, the main idea of C&W $L_2$ attack is to minimize the "perturbation" $z$ through an optimizer. In addition, the most important contribution in this algorithm is that they successfully translate the adversarial goal of misclassification to a mathematical formula, which in Equation (5) is denoted as the function $f$.

In the following part of the experiments, we mostly generate adversarial examples in this method. The clean images $x$ are all chosen from the test dataset. We define the hyper-parameter $\mathcal{K}$ as 10 because it has the best effect of our attacks. As for $c$, as is described in the original paper, the algorithm uses Binary Search to find the best value automatically. As a whole, we only give the clean image $x$ and the hyper-parameter $\mathcal{K}$, and the algorithm will begin to minimize the objective function. The initial value of variable $z$ is a zero vector that has the same shape with $x$. Finally, we get the adversarial examples $\frac{1}{2}(\tanh(z) + 1)$ according to the best value $z$.

These works about substitute training have achieved a high success rate in non-targeted black-box attacks. However, the weakest point is that they don't limit the size of perturbations, and the adversarial examples they craft are mostly unrecognized and meaningless. Thus, there's a big space for promoting these kinds of attacks. In addition, the query times of their methods are huge and unacceptable in some situations.

## 4. Approaches

In this section, we will show the main process of how we overcome some shortcomings of previous works and conduct some new works that previous works didn't perform like targeted attacks. This section will partly include two parts that respectively introduce how we improve the Jacobian-based Augmentation and the strategy that boosts the black-box attacks.

### 4.1. Gradient-Based Linear Dataset Augmentation Method

After careful research about the Jacobian-based Augmentation, we found that the accuracy of their substitute model is not very high, and the targeted attacks won't usually be successful. Intuitively, we assume that increasing the test accuracy of substitute models is a feasible way to improve the success rate of targeted attacks. In addition, by visualizing the augmenting process of Jacobian-based Augmentation, we found that the augmented images this method crafts are unrecognizable to humans' eyes after a few epochs because they repeatedly augment the images newly crafted in the last epoch, which results in the training dataset of substitute models going far away from the original dataset. Consequently, the accuracy of substitute models decreases due to these unrecognizable images. In addition, the query times of target model are mostly useless.

Due to these discoveries, we propose the **Gradient-Based Linear Dataset Augmentation Method** that augments the initial dataset for a series of epochs with the gradient information of substitute models, and, every epoch, we only augment the initial dataset, different from the Jacobian-based augmentation [15] from the augmented data of last epoch. We found that the accuracy of substitute models increase a lot because the new augmented images are all from the initial images, so they have a better visual effect than the Jacobian-based augmentation.

In Figure 2, we can quickly know that the augmenting dataset our method obtains is much smaller than the Jacobian-based Augmentation so that we can control our augmenting epochs and the query times freely.

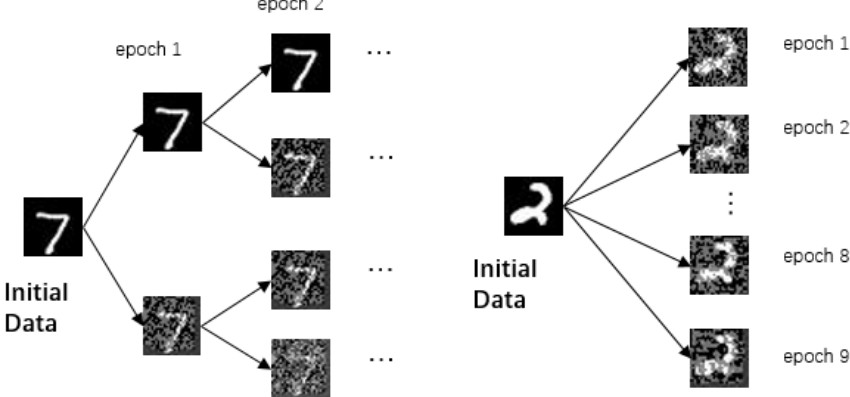

**Figure 2.** Improvement to Jacobian-based Augmentation. On the left is the original Jacobian-based Augmentation flowchart, and the number of images is increasing exponentially with the epoch growing and the query times will be unacceptable after a few epochs. The right pattern is our improvement that all the augmenting images are from the initial data, and the query times increase *linearly*.

Corresponding to the augmenting function in Section 3, we can describe our augmenting method as this:

$$S_{\rho+1} = \{x + \lambda \cdot \text{sgn}(\nabla_x[\tilde{F}(x, \theta_\rho)]) : x \in S_0\}, \tag{7}$$

where $\lambda$ is the hyper-parameter that determines the step size of each epoch, $S_0$ is the initial dataset, $F$ is the substitute model, $\theta_\rho$ is $F$'s parameters that changes with the epoch $\rho$, *sgn* is the sign function and the $\nabla_x[\tilde{F}(x, \theta_\rho)]$ means the gradient of substitute model $F$'s output to its input $x$. Here, we can find that the we just choose the direction towards maximizing the complexity of substitute model $F$'s training dataset via the *gradient information*. This is why we called our method the Gradient-Based Linear Dataset Augmentation Method.

### 4.2. Ensemble Training

During the studying of the white-box attacking method C&W [11], we found that the ensemble attack [19–21] is highly efficient and effective for increasing the transferability of adversarial examples. Thus, we try to train several substitute models $F_1, F_2, ..., F_N$ to generate our ensemble substitute models' attacks. However, despite the increase of attacking success rate, there's a serious problem that the query times of training substitute models are linearly increasing with the number of substitute models, which is not what we expect.

Inspired by the thought of ensemble attack that combines the outputs of several models' logits layers [19], we try to combine the gradient information of several substitute models during augmentation:

$$S_{\rho+1} = \{x + \lambda \cdot \text{sgn}(\sum_{i=1}^{N} \omega_i \nabla_x[\tilde{F}(x, \theta_\rho)]) : x \in S_0\}, \tag{8}$$

where $\omega_i(i = 1, 2, ..., N)$ is the weight of $F_i$'s gradient. In addition, in this way, we elevate the success rate of the gradient-based linear augmentation a lot. The detail results will be presented in Section 6.

Combining these two ways above, we have our final process in Algorithm 1.

---

**Algorithm 1** A gradient-based linear dataset augmentation method for ensemble training.

---

**Input:**    The target black-box model $O$; The maximum of iterative epoch $\rho_{max}$; Initial dataset $S_0$ for augmentation; Step size parameter $\lambda$; Number of substitute models $N$; Weights of subtitute models' gradients $\omega_1, \omega_2, ..., \omega_N$.

**Output:**    Parameter of trained substitute models $\theta_{F_1}, \theta_{F_2}, ..., \theta_{F_N}$

1:  Define Architecture $F_1, F_2, ..., F_N$

2:  $\rho \leftarrow 0, S_\rho \leftarrow S_0$

3:  **for** $\rho \in 0, 1, ..., \rho_{\max}$ **do**

4:       $D \leftarrow \{(x, O(x)), x \in S_\rho\}$

5:      **for** $i \in 1, 2, ..., N$ **do**

6:           $\theta_{F_i} \leftarrow train(F_i, D)$

7:      **end for**

8:      **if** $\rho < \rho_{\max}$ **then**

9:           //Ensure all augmented data used to train substitute models

10:          $S_{\rho+1} = \{x + \lambda \cdot \text{sgn}(\sum\limits_{i=1}^{N} \omega_i \nabla_x[\tilde{F}(x, \theta_\rho)]) : x \in S_0\}$

11:     **end if**

12: **end for**

13: **return** $\theta_{F_1}, \theta_{F_2}, ..., \theta_{F_N}$

---

## 5. Evaluation

When generating these kinds of experiments, the first step we need to take is training the substitute models. In addition, to be specific, we should choose the architecture of substitute models and the dataset augmentation method. In addition, we will prove that the augmentation method plays a much more important role than the architecture in this section. The previous sections have introduced two methods for generating high-transferability adversarial examples via training substitute models. In addition, these two methods are not independent and can be combined to generate a better effect. In the next two subsections, we will present our experimental results and evaluate our methods.

### 5.1. Setup

#### 5.1.1. Datasets

We use MNIST [22] and GTSRB [23] datasets to generate our experiments. MNIST is a well-known grayscale handwritten digits dataset that contains 60,000 training images and 10,000 test images of 10 classes, and in the experiments we also keep its scale as $28 \times 28$. Another dataset is the German Traffic Sign Recognition Benchmark, which contains 43 classes of 39,209 training and 12,630 test images. Unlike the MNIST dataset that only has only channel, GTSRB is a colorful dataset with R(red),G(green) and B(blue) channels. In the experiment we rescale all images of GTSRB into $48 \times 48$. All of the image pixels are preprocessing to range $(-0.5, 0.5)$ [11] for the convenience of using the C&W attack.

#### 5.1.2. Initial Dataset

The initial data $S_0$ are all randomly picked from the test data, and the number is 600 and 300, which are all ensured no more than 1% of the training datasets.

#### 5.1.3. Model Architectures Selection

In our experiments, we select the CNN model that behaves efficiently in [24] as our black-box model $O$. In addition, after training 10 epochs, the test accuracies on MNIST and GTSRB are up to

98.93% and 96.27%. We save these two models locally as our target models. The architectures of substitute models come from [11] and are adjusted properly. Finally, we choose five architectures A~E to define our substitute models. The architecture of these models is shown in Table 1.

**Table 1.** Table of the architectures of substitute models. Here, the Conv and Relu are convolution layer and the Relu activation function, FC means the fully connected layer, and Softmax is the last layer of CNN classification model. Notice that the NC in Softmax layers is the number of the classes that the target dataset contains.

| A | B | C | D | E |
|---|---|---|---|---|
| Conv(32,3)+Relu | Conv(32,3)+Relu | Conv(32,3)+Relu | Conv(32,3)+Relu | Conv(32,3)+Relu |
| Conv(32,3)+Relu | MaxPooling(2) | Conv(32,3)+Relu | Conv(32,3)+Relu | MaxPooling(2) |
| MaxPooling(2) | Conv(64,3)+Relu | MaxPooling(2) | MaxPooling(2,2) | Conv(64,3)+Relu |
| Conv(64,3)+Relu | Conv(64,3)+Relu | Conv(64,3)+Relu | Conv(64,3)+Relu | MaxPooling(2) |
| Conv(64,3)+Relu | MaxPooling(2) | MaxPooling(2) | Conv(64,3)+Relu | FC(200)+Relu |
| MaxPooling(2) | FC(200)+Relu | FC(200)+Relu | MaxPooling(2) | FC(NC)+Softmax |
| FC(200)+Relu | FC(200)+Relu | FC(200)+Relu | FC(200)+Relu | |
| FC(200)+Relu | FC(NC)+Softmax | FC(NC)+Softmax | FC(NC)+Softmax | |
| FC(NC)+Softmax | | | | |

### 5.1.4. Hyper-Parameters

We use the Stochastic Gradient Descent(SGD) optimizer with the learning rate 0.01, decay rate $10^{-6}$ and the momentum 0.9 for training all the substitute models. In addition, the loss function is the cross-entropy function. The hyper-parameter $\lambda$ in Algorithm 1 is 0.4 according to our several tests. As for the weight vectors $\omega_i$, because we have no prior knowledge for which architecture plays a more important role, we just set them the same, and the value is 1 divided by the number of the substitute models. In addition, in the final part of this section, we will find that these architectures mostly perform the same separately so that setting $\omega_i$ the same is reasonable.

### 5.1.5. Experiment Environment

All our experiments are generated via Keras using TensorFlow as backend on Windows 10 (Microsoft, Redmond, WA, USA). In addition, our experiment equipment is one Nvidia GeForce GTX 1080 GPU (Santa Clara, CA, USA) with 8 GB video memory and the Intel E5 CPU (Santa Clara, CA, USA) with a memory of 32 GB.

### 5.2. Query Times

The first effect we want to know is the decrease in query times. In addition, as the process of our method, we only augment the initial image set in each substitute training epoch, so that the increasing number of query times is just the same as the number of initial images. In other words, the query time increases with the substitute training epoch *linearly*. Moreover, speed is another piece of information that we want to compare with the Jacobian-based augmentation method.

Figure 3 shows the increase of query times and the time cost with the substitute epoch $\rho$. The increasing rate of query times changes from exponentially to linearly. In addition, we can find that the time cost with the development of augmentation nearly stays the same in our augmentation while the Jacobian-based Augmentation grows dramatically. For that reason, we called our augmentation as Linear Augmentation in comparison with the exponential growth of Jacobian-based Augmentation.

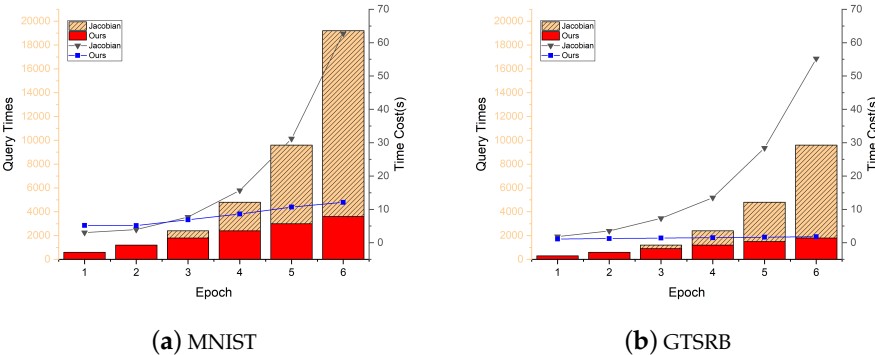

(**a**) MNIST      (**b**) GTSRB

**Figure 3.** Increase of query times and time cost with the substitute epoch. The column bars indicate the query times and the lines with symbols describe the time cost of each epoch.

### 5.3. Accuracy of Substitute Models

During the design of augmentation, we talked about the problem of visual effect that directly influences the accuracy of substitute models. In addition, at the beginning of this part, we declare that all the accuracies are tested on the testing set of the dataset and it is defined as:

$$accuracy = \sum_{i \in T - S_0} I(F(x_i) = y_i), \tag{9}$$

where $T$ is the test dataset and $y_i$ is the ground truth label of $x_i$, $I$ is the indicator function.

#### 5.3.1. Augmentation without Ensemble Training

As expected, the accuracy of our augmentation will increase faster than the Jacobian-based augmentation. Figure 4 shows the result of accuracy enhancement of our improvement. As the figure shows, our method performs much better than the Jacobian-based models. For the MNIST dataset, the accuracy of our method achieves more than 95% while the Jacobian-based Augmentation can only achieve 80%. In addition, in the experiment of GTSRB, we can find that the accuracy of Jacobian-based Augmentation can only achieve half of our method.

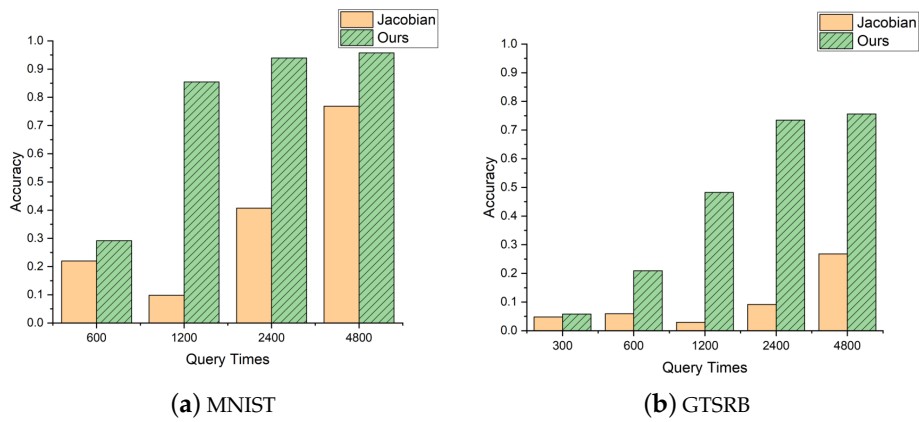

(**a**) MNIST      (**b**) GTSRB

**Figure 4.** Accuracy of substitute models changes with the query times.

We also analyze the reasons for this phenomenon. On the one hand, the images augmented in our method have better visual effects, which means that they are closer to the real images and can be recognized by humans more easily. In other words, a lot of fuzzy images mislead the substitute model and indirectly decrease its accuracy. On the other hand, the training information we use is the

confidence vectors instead of the label, which avoid the misleading images used as the incorrect labels during the process of substitute training. For example, there is an image of MNIST that Jacobian-based augmentation crafts after a few epochs, and it is quite unrecognizable to human eyes. However, during the substitute training, the Jacobian-based augmentation that gives the probability of 1.0 is that this image is some certain figure of 0~9, while our substitute training set may only consider this fuzzy image a certain number with the probability of 0.1 to 0.5 rather than 1. This kind of misleading may be another reason that Jacobian-based augmentation cannot obtain high accuracy.

### 5.3.2. Ensemble Training

After evaluating the improvement of our augmentation method to the accuracy, we continue to evaluate the influence of ensemble training on the accuracy of substitute models. In this part, we train five substitute models with ensemble training, and the experiment environment remains the same as the previous. Table 2 shows the influence on the accuracy of substitute models A~E.

**Table 2.** Accuracy of the substitute models via ensemble training. Compared with Figure 4, we can find that our ensemble training method doesn't decrease the accuracy of substitute models. Therefore, the ensemble attack is feasible through these models according to the analysis of the importance of the substitute models' accuracy.

| Dataset | Query Times | Substitute Model Architecture | | | | | Mean Accuracy |
|---------|-------------|--------|--------|--------|--------|--------|---------------|
|         |             | A | B | C | D | E | |
| MNIST | 600 | 86.83% | 85.88% | 85.09% | 83.61% | 87.64% | 85.81% |
|       | 1200 | 91.16% | 90.91% | 90.89% | 92.81% | 91.78% | 91.51% |
|       | 2400 | 96.00% | 95.89% | 95.14% | 96.39% | 95.13% | 95.71% |
|       | 4800 | 97.33% | 96.85% | 96.47% | 97.29% | 96.31% | 96.85% |
| GTSRB | 300 | 11.14% | 5.72% | 8.20% | 6.48% | 5.24% | 7.36% |
|       | 600 | 22.20% | 8.71% | 9.58% | 9.51% | 7.62% | 11.52% |
|       | 1200 | 42.38% | 32.73% | 33.62% | 40.42% | 41.87% | 38.20% |
|       | 2400 | 67.98% | 74.07% | 75.21% | 64.39% | 75.62% | 71.45% |
|       | 4800 | 75.59% | 79.52% | 82.14% | 74.36% | 78.99% | 78.12% |

### *5.4. Success Rate and Distortion*

In this subsection, we introduce the ultimate goal of our training substitute models.

For any attack method, the success rate is the first and the most important metric. In our method, we define our success rate as

$$success\_rate = \frac{\sum_{i \in S} I(O(x_i) = t_i, F(x_i) = t_i)}{\sum_{i \in S} I(F(x_i) = t_i)}, \tag{10}$$

where $S$ is the test dataset for crafting adversarial examples, $t$ is the target class, and $I$ is the indicator function. The meaning of the definition is the number of adversarial examples that can be misclassified to the target class both by the substitute model $F$ and the target model $O$ divided by the number of adversarial examples that can be misclassified by the substitute models. In addition, we also evaluate our adversarial examples by the $L_2$ distance of perturbation, which is commonly used and we don't repeat its definition here again.

### 5.4.1. Single Model Attack

In this part, we generally compare the effect of our method with the Jacobian-based Augmentation attack. Firstly, we generate the FGSM attack [6] implemented in the paper of Jacobian-based Augmentation [15]; here, we set the hyper-parameter $\varepsilon$ 0.2 and craft adversarial examples using

the substitute models we trained previously. In addition, we get the result shown in Table 3 that our attack has better performance in the success rate when the perturbation stays the same.

**Table 3.** Effect of non-targeted attack.

| Dataset | Success Rate | | $L_2$ Distortion | |
|---|---|---|---|---|
| | Jacobian | Ours | Jacobian | Ours |
| MNIST | 5.80% | 24.55% | 3.83 | 3.84 |
| GTSRB | 9.48% | 32.66% | 2.37 | 2.37 |

After simply comparing the non-targeted result, we continue to generate the targeted adversarial examples via a C&W attack method. Table 4 shows $L_2$ distance of perturbation and the success rate. As the table shows, Jacobian-based Augmentation fails in generating a targeted attack. It is explainable that Jacobian-based augmentation failed here. Given an adversarial example $x^*$ that substitute models successfully misclassify as the target class $t$, the target model may recognize $x^*$ as another class because of the low accuracy of the substitute model.

**Table 4.** Effect of targeted attack.

| Dataset | Success Rate | | $L_2$ Distortion | |
|---|---|---|---|---|
| | Jacobian | Ours | Jacobian | Ours |
| MNIST | 0 | 56.67% | – | 3.87 |
| GTSRB | 0 | 36.67% | – | 3.13 |

### 5.4.2. Ensemble Training and Attacks

In this part, we generate our Linear Augmentation with the Ensemble Training Method and craft adversarial examples with multiple substitute models. For better descriptions of our methods' boosting effect, we only introduce the targeted attacks in our approaches.

Figure 5 shows the final effect of our black-box attack on the MNIST and GTSRB datasets. In addition, digitally, we attack the MNIST classification 90 model with the success rate of 97.7% and the GTSRB with 92.78% success rate. Those two groups of pictures are randomly picked from our attack results. The adversarial examples all look like the original images and won't be recognized incorrectly by human vision, but the target model will misclassify them as the target classes.

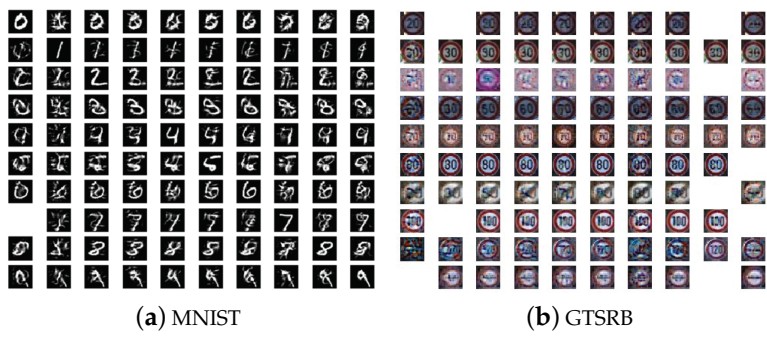

(**a**) MNIST        (**b**) GTSRB

**Figure 5.** Final adversarial examples of our attacks. The diagonal contains the original images, and for every column image; they are classified into the same target class but the vision stays like the original image. The empty area indicates the failure of transferable adversarial examples.

In addition, in detail, we analyze the final experimental data in Figure 6. As is shown in the figure, with the increase of the number of substitute models, the success rate grows fast. On the other hand, the distortion increases with the substitute models, which is not good news for us. Thus, how to

balance success rate and distortion is a vital problem, and we should choose the proper number of substitute models according to the situation.

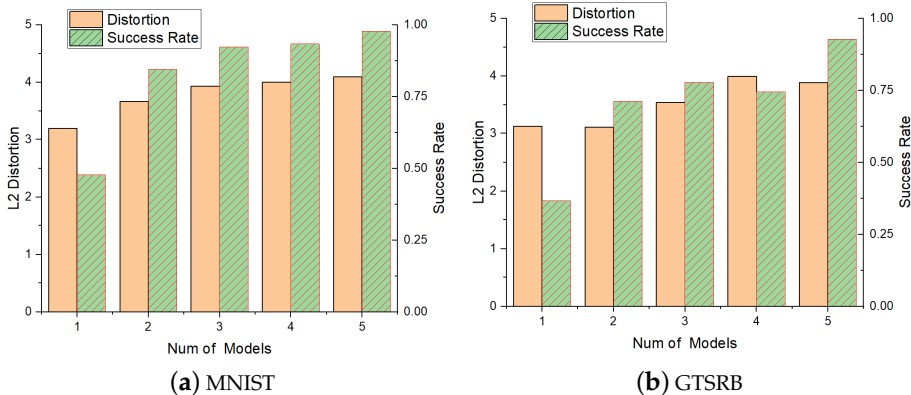

**Figure 6.** Influence of number of substitute models. The attacking success rate increases quickly when the number of substitute models increases.

*5.5. Extension*

Finally, we also research the influence of different architectures on the effect of our adversarial attacks. In addition, we take the MNIST as experiment material to generate the single model attack. The results of this experiment are shown in Table 5. Through the table, we can find that there's nearly no side influence for different architectures. Thus, corresponding to the previous statement, selecting the first DNN architecture has little influence on the analysis of querying times and the time cost.

**Table 5.** Influence of different architectures on the final results. Here, 'SR' means the success rate.

| DNN | A | B | C | D | E |
|-----|------|------|------|------|------|
| SR | 56.67% | 50.00% | 48.31% | 53.33% | 44.44% |
| $L_2$ | 3.87 | 3.65 | 3.96 | 3.39 | 3.82 |

## 6. Conclusions

In this paper, we succeed at generating a high success rate targeted black-box attack via training multiple local substitute models with the Gradient-Based Augmentation Method. Through the research of Jacobian-based Augmentation by Papernot et al. [15], we come up with the improvement of augmenting the data from initial data for every epoch to highly increase the accuracy of substitute models. On the other side, we can avoid the exponential growth of query times of the original Jacobian-based Augmentation. In this way, we boost the state-of-the-art targeted black-box attack via substitute training a lot. Thus, we come to the conclusion that high accuracy of substitute models is a necessary condition to generate the targeted black-box attack via substitute training. Furthermore, we transfer the idea of ensemble attacks to our process of training substitute models and propose the Ensemble Training Method, which effectively increases the success rate. Finally, we successfully attack the MNIST and GTSRB image recognition system with over a 90% success rate. Furthermore, in theory, we can train as many substitute models as we want to generate ensemble attacks without increasing the query times of a target model.

In conclusion, training local substitute models is an effective and efficient way to attack target black-box image classification. The main advantage is that, for a special system, once the substitute models have been trained, we can craft the adversarial examples for any number of images without any extra queries. Furthermore, in the future, when we have difficulty promoting the success rate through adjusting the methods for crafting adversarial examples, trying to increase the accuracy of substitute models through querying may be another direction that may work better. In addition, the

ensemble training and attacks are highly efficient approaches to promoting the adversarial success rate. We can take advantage of them properly as a side way to improve the effects of adversarial attacks.

**Author Contributions:** Conceptualization, Y.-a.T.; Data curation, H.J.; Formal analysis, Y.-a.T.; Funding acquisition, Y.-a.T.; Investigation, X.G.; Methodology, X.G.; Project administration, X.K.; Resources, H.J.; Software, Q.Z.; Supervision, X.K.; Validation, Q.Z.; Writing—review and editing, X.G.

**Funding:** This work was supported by the National Natural Science Foundation of China under Grant No. 61876019.

**Conflicts of Interest:** The authors declare that there is no conflict of interests regarding the publication of this article.

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
