# Peer review of "Boosting Targeted Black-Box Attacks via Ensemble Substitute Training and Linear Augmentation"

_applsci, doi:10.3390/app9112286_

Round 1

Reviewer 1 Report

Please consider the following changes/improvements:

rerun the latex version, publication numbers are not shown.

Explain always when the acronym is referred for first time (e.g. SVM line 33)

Related work section should be rewritten from scratch focusing on previous works conducted with a closing paragraph including a comparison paragraph highlighting the authors' contribution.

enumerate functions

enlarge figures and pictures

Author Response

(1) Rerun the latex version, publication numbers are not shown.

Response:

We’re sorry for the carelessness of not checking for this problem in the PDF file, and the next version of the paper will include this.

(2) Explain always when the acronym is referred for the first time.

Response:

Thanks for your suggestion. Based on your comments, we have made the corrections and highlight it mostly in section I.

(3) Related Work section should be rewritten from scratch focusing on previous works conducted with a closing paragraph including a paragraph highlighting the author’s contribution.

Response:

We really appreciate this suggestion. We have realized this problem that Related Work shouldn’t contain too much irrelevant content. Combining another reviewer’s advice and yours, we have rewritten this section and change the location of some content previously in this section. The detailed adjustments are:

i.                     We added one chapter named Threat Model before Related Work for introducing the threat models that most papers about adversary will introduce and some notations that may be used later.

ii.                   We have rewritten the section Related Work thoroughly including a closing paragraph indicating the weakness of previous works and what we can do in our paper.

(4)(5) Enumerate functions and Enlarge figures and pictures

Response:

We have noticed that the format of the enumerate function have some problems and changed it.

Thanks for your reminding and we have enlarged some figures that are too small.

Author Response

(1) There are so many English grammar errors.

Response:

We have carefully checked our words and sentences again, and we are very sorry for dyslexia that these problems take to you.

(2) Generally, in “Introduction section”, the papers related with your topics are introduced with some disadvantage. You just suggested your method very simplely, which improves the existing problem(s) and explains contribution. However, you explain your proposal detaily and conclude with two examples. These conclusions should be presented in the conclusion section or in an experimental section. And also, the second paragraph in contribution (line 89- 90 in page 3) should be in the experimental section (analysis part) in your paper.

Response:

Thank you for your careful reading and nice comments. We have merged some paragraphs in the Introduction section and simplified the introduction of our proposals, which you may see in the next version of manuscripts in the first section highlighted with yellow color. Besides, the examples of our experiments have been moved to the Evaluation section as your advice.

(3) At the end of introduction section, there should be a paragraph which explains the construction of your paper.

Response:

Thanks for your nice suggestion. We have added one paragraph that includes the main construction of our paper.

(4) What is “w” in the line 119 of page 3 “search for w that solve.”?

Response:

We are sorry for our carelessness of forgetting changing the “w” in the original paper of C&W attack into “z” in our paper because of the different notations in these two papers.

The meaning of “z” will be answered in the next question.

(5) What is “z” in the equation (? – put the equation number for all equations) in line 119 of page 3?

Response:

We are sorry for taking you this confuse. We have introduced this equation more specifically in Section 3.2. And the meaning of the parameter “z” will be explained in the last paragraph of this part. Generally, we can say “z” presents the size of the perturbations.

Moreover, as your advice, we have put the equation number for all equations.

(6) What is meant for notation “?(?′)?”? This notation is not explained yet.

Response:

We are sorry for our carelessness. For avoiding other problems like this and corresponding to another advice for rewrite Section 2, we have added one Section names Threat Model to introduce the threat model and notations. “?(?′)?” here means the i-th component of the output vector “?(?′)”.

(7) In section 2 (Related Works), only very related works with your idea (work) should be presented in detail. However, you described your main idea in this section. Rewrite whole section here with related work such as Jacobian-based augmentation and so on.

Response:

We really appreciate this suggestion. We have realized this problem that Related Work shouldn’t contain too much irrelevant content. Combining another reviewer’s advice and yours, we have rewritten this section and change the location of some content previously in this section. The detailed adjustments are:

i.                     We added one chapter named Threat Model before Related Work for introducing the threat models that most papers about adversary will introduce and some notations that may be used later.

ii.                   We rewrite the Section Related Work thoroughly that only includes very related works.

(8) The whole description in section 2 should be combined with your section 3. The combined section is deserved for your proposed method.

Response:

Thanks very much for your suggestion. We have rewritten the Related Work section and add some content for combining these two parts.

(9) You should explain the equation described in the line 119 of page 3 in detail. It is the most important idea in your paper.

Response:

As we respond previously, we have added some content for describing this method. And here we declare again that it’s not the method we propose and we just use it to craft adversarial examples.

(10) How do you determine the weight value in equation described in the line 155 of page 5 before using them? In the experiment (line 179 of page 6), you decide the value as the mean value of the number of the model. Why did you do?

Response:

Thanks for your nice question. We have added these words highlighted in Section 5.1:

“Because we have no prior knowledge that which architecture plays a more important role, we just set them the same. And in the final part of this section, we will find that these architectures mostly perform the same separately so that we set the weight values the same is reasonable.”

(11) Where is your reference list ?

Response:

We’re sorry for the carelessness of not checking for this problem in the PDF file, and the next version of the paper will include this.

Round 2

Author Response

Point 1: Line 95 of page 3: Why did you restrict that 10% adversary ? Give the reason (you just describe closer to the real case).

Response 1: Thanks for your careful reading. We restrict that 10% adversary mainly because the previous works in this way mostly exceed this limit due to their exponentially increasing. Besides, our method can achieve the best effect only in this limit.

Point 2: What is the range of x in equation (3) in the page 3?

Response 2:  As we describe in the Setup section, we preprocess all the image pixel in range(-0.5, 0.5). However, for different models this range may be different so we don’t bother to describe the range of x either here nor in the algorithm. But we do clip the value of each pixel of x during every augmentation epoch.

Point 3&4: Line 110 of page 4, the notation x∈??+1 is incorrect. Correct it as x∈??+1.

Delete “,” in equation (6). You copied this equation from page 9 of the reference paper [11] exactly including “,”.

Response 3: We are very sorry for the carelessness, and thanks to your attention very much. We have changed the errors in our paper.